# Learning Symbolic Persistent Macro-Actions for POMDP Solving Over Time

**Celeste Veronese**[*]                                    CELESTE.VERONESE@UNIVR.IT
**Daniele Meli**[*]                                          DANIELE.MELI@UNIVR.IT
**Alessandro Farinelli**                            ALESSANDRO.FARINELLI@UNIVR.IT
*Dept. of Computer Science, University of Verona, Italy*

**Editors:** Leilani H. Gilpin, Eleonora Giunchiglia, Pascal Hitzler, and Emile van Krieken

## Abstract

This paper proposes an integration of temporal logical reasoning and Partially Observable Markov Decision Processes (POMDPs) to achieve interpretable decision-making under uncertainty with macro-actions. Our method leverages a fragment of Linear Temporal Logic (LTL) based on Event Calculus (EC) to generate *persistent* (i.e., constant) macro-actions, which guide Monte Carlo Tree Search (MCTS)-based POMDP solvers over a time horizon, significantly reducing inference time while ensuring robust performance. Such macro-actions are learnt via Inductive Logic Programming (ILP) from a few traces of execution (belief-action pairs), thus eliminating the need for manually designed heuristics and requiring only the specification of the POMDP transition model. In the Pocman and Rocksample benchmark scenarios, our learned macro-actions demonstrate increased expressiveness and generality when compared to time-independent heuristics, indeed offering substantial computational efficiency improvements.

## 1. Introduction

In complex and uncertain decision-making, efficiently handling large action spaces and long planning horizons is still a major challenge. Most popular and effective approaches to online solving Partially Observable Markov Decision Processes (POMDPs, Kaelbling et al. (1998)), e.g., Partially Observable Monte Carlo Planning (POMCP) by Silver and Veness (2010) and Determinized Sparse Partially Observable Tree (DESPOT) by Ye et al. (2017), rely on Monte Carlo Tree Search (MCTS). These approaches are based on online simulations performed in a *simulation environment* (i.e. a black-box twin of the real POMDP environment) and estimate the value of actions. However, they require domain-specific *policy heuristics*, suggesting best actions at each state, for efficient exploration. Macro-actions (He et al. (2011); Bertolucci et al. (2021)) are popular policy heuristics that are particularly efficient for long planning horizons. A macro-action is essentially a sequence of suggested actions from a given state that can effectively guide the simulation phase towards actions with high utilities. However, such heuristics are heavily dependent on domain features and are typically handcrafted for each specific domain. Defining these heuristics is an arduous process that requires significant domain knowledge, especially in complex domains. An alternative approach, like the one by Cai and Hsu (2022), is to learn such heuristics via neural networks, which are, however, uninterpretable and data-inefficient.

---

[*] Equal contribution. Corresponding authors.

This paper contributes to neuro-symbolic AI, within the informed machine learning field, by extending the methodology proposed by Meli et al. (2024) to learn Event Calculus (EC, Kowalski and Sergot (1989)) theories using Inductive Logic Programming (ILP, Muggleton (1991)) in the Answer Set Programming (ASP, Lifschitz (1999)) formalism. We show that *persistent* macro-actions can be learned and leveraged to guide exploration toward high-value policy subtrees in MCTS-based planners such as POMCP and DESPOT. Introducing temporal abstraction enables a more faithful modeling of POMDP dynamics, enhancing the expressiveness of learned policy heuristics while also improving computational efficiency.

## 2. Related Work

As shown by Kim et al. (2019), handcrafting task-specific macro-actions for efficient POMDP planning requires significant domain knowledge. Inspired by the success of AlphaZero and its variants (Silver et al. (2018)) to guide MCTS exploration with neural networks, Lee et al. (2020) combine online planning and learning with macro-actions in DESPOT with a recurrent actor-critic architecture. Similarly, Subramanian et al. (2022) exploit a recurrent autoencoder solution for training a rocksample agent via Reinforcement Learning (RL). However, a major limitation of black-box methods, such as neural networks, is the large amount of data and time required for training, as well as the limited interpretability and generalization out of the training setting. Recent research is then moving towards merging (PO)MDP planning with symbolic (logical) reasoning, with the potential to increase interpretability and trust (Maliah et al. (2022); Mazzi et al. (2023a)). In particular, for long-horizon tasks, Linear Temporal Logic (LTL) has been used by De Giacomo et al. (2019) to define additional reward signals for MDP agents, or by Leonetti et al. (2012) to define bounds on temporal action sequences. However, these methods rely on significant prior knowledge that may not be readily available in complex real-world domains. Differently, De Giacomo et al. (2020) learn LTL specifications for reward shaping from MDP traces. However, they assume full observability. Moreover, similar to neural networks by Cai and Hsu (2022), bad training data may lead to bad reward signals, with a significant performance drop. On the contrary, biasing MCTS exploration as in Mazzi et al. (2023b); Meli et al. (2024); Gabor et al. (2019) was proven more robust to bad heuristics.

Differently from Gabor et al. (2019), we propose to learn EC action theories in the ASP formalism (taking inspiration from Erdem and Patoglu (2018); Tagliabue et al. (2022)), from traces of execution of the agent in easy-to-solve scenarios. To this aim, we rely on ILP, which has already been successfully used to explain black-box models by D'Asaro et al. (2020); Veronese et al. (2023), and to generate policy heuristics that can be exploited to improve RL performances by Furelos-Blanco et al. (2021); Veronese et al. (2025). Our work represents an extension of the methodology proposed by Meli et al. (2024), in which only *time-independent* ASP action theories were learned, limiting the expressiveness of the generated axioms and, consequently, their impact in improving planning performances.

## 3. Background

We now introduce the main definitions for POMDPs, ASP and ILP, and our domains to exemplify the theoretical aspects.

### 3.1. Partially Observable Markov Decision Processes (POMDPs)

A POMDP (Kaelbling et al. (1998)) is a tuple $(S, A, O, T, Z, R, \gamma)$, where $S$ represents a set of partially observable *states*, $A$ stands for a set of *actions*, $Z$ denotes a finite set of *observations*, $T : S \times A \to \Pi(S)$ functions as the state-transition model with probability distribution $\mathcal{B} = \Pi(S)$, also known as *belief*, over states. Additionally, $O : S \times A \to \Pi(Z)$ operates as the observation model, whilst $R$ denotes the reward function with discount factor $\gamma \in [0, 1]$. The agent's objective is to compute a policy $\pi : \mathcal{B} \to A$ that maximizes the discounted return $E[\sum_{t=0}^{\infty} \gamma^t R(s_t, a_t)]$.

#### 3.1.1. Online POMDP planning

We consider two online POMDP solvers, POMCP by Silver and Veness (2010) and DESPOT by Wu et al. (2021). Both solutions rely on MCTS combined with a particle filter to approximate the belief distribution as a set of particles, i.e., state realisations. MCTS performs online simulations (in a black-box twin of the real POMDP environment) from the current belief, to estimate the value of actions. Specifically, at each time step of execution, it builds a state-action tree, selecting actions in $A$ and propagating particles according to the transition model. It then records the history $h$ of already explored state-action pairs. Given a maximum number of simulations (corresponding to particle-action sequences), in POMCP the best action at a belief node is selected according to UCT (Kocsis and Szepesvári (2006)), which maximizes $V_{UCT}(ha) = V(ha) + c \cdot \sqrt{\frac{\log N(ha)}{N(h)}}$, being $V(ha)$ the expected return achieved by selecting action $a$, $N(h)$ the number of simulations performed from history $h$, $N(ha)$ the number of simulations performed while selecting action $a$, and $c$ the exploration constant. If no history is available (leaf nodes in MCTS tree), a rollout policy (typically random) is used to select the next action. In DESPOT, the exploration is guided by a lower and an upper bound ($l(b)$ and $u(b)$, respectively) on the root belief node $b$, denoting the minimum and maximum expected value for it. The lower bound is usually computed as the value associated with a default action, similar to rollout action in POMCP. DESPOT then explores only subtrees with value between the bounds; thus, if the bounds are accurately defined according to domain-specific heuristics, the solver requires fewer simulations than POMCP. If $u(b) - l(b) < \varepsilon \in \mathbb{R}^+$, the default action is directly applied.

### 3.2. Answer Set Programming and Event Calculus

Answer Set Programming (ASP) by Lifschitz (1999) represents a planning domain by a sorted signature $\mathcal{D}$, with a hierarchy of symbols defining the alphabet of the domain (variables, constants and predicates). Logical axioms or rules are then built on $\mathcal{D}$. In this paper, we exploit the ASP formalism to represent knowledge about the MDP. To this aim, we consider normal rules, i.e., axioms in the form $\mathtt{h} : -\mathtt{b_1}, \ldots, \mathtt{b_n}$, where the body of the rule (i.e. the logical conjunction of the literals $\mathtt{b_1} \wedge \cdots \wedge \mathtt{b_n}$) serves as the precondition for the head $\mathtt{h}$. In our setting, body literals represent the state of the environment, while the head literal represents an action. Given an ASP problem formulation $P$, an ASP solver computes the *answer sets*, i.e., the minimal models satisfying ASP axioms, after all variables have been *grounded* (i.e., assigned with constant values). Answer sets lie in Herbrand base $\mathcal{H}(P)$,

defining the set of all possible ground terms that can be formed. In our setting, answer sets contain the feasible actions available to the agent.

ASP allows the representation of action theories accounting for the temporal dimension, introducing an ad-hoc variable `t` for representing the discrete time step of execution. In this paper, we focus on representing and learning EC theories in the ASP formalism. EC (Kowalski and Sergot (1989)) is designed to model events and their consequences within a logic programming framework. This is done relying on the following inertial axioms:

$$\text{holds(F,t) :- init(F,t).} \tag{1}$$
$$\text{holds(F,t) :- holds(F,t-1), not end(F,t).}$$

meaning that an atom `F` starts holding when an `init` event occurs (i.e., the atom is ground), until `end` event does not occur. In this paper, we reformulate Equation (1) using `contd(F,t)` in place of `not end(F,t)`, i.e., representing the condition for `F` to keep holding.

### 3.3. Inductive Logic Programming

An ILP problem $\mathcal{T}$ under ASP semantics is defined as the tuple $\mathcal{T} = \langle B, S_M, E \rangle$, where $B$ is the *background knowledge*, i.e. a set of known atoms and axioms in ASP syntax (e.g., ranges of variables); $S_M$ is the *search space*, i.e. the set of all possible ASP axioms that can be learned; and $E$ is a set of *examples* (e.g., a set of ground atoms constructed, in our case, from traces of execution). The goal is to find an *hypothesis* $H \subseteq S_M$ such that $B \cup H \models E$, where $\models$ denotes entailment. For this purpose, we employ the ILASP learner by Law (2023), wherein examples are *Context-Dependent Partial Interpretations* (CDPIs), i.e., tuples of the form $\langle e, C \rangle$. $e = \langle e^{inc}, e^{exc} \rangle$ is a partial interpretation of an ASP program $P$, such that $e^{inc} \subseteq \mathcal{H}(P)$, $e^{exc} \nsubseteq \mathcal{H}(P)$; $C$ is a set of ground atoms called *context*. The goal of ILASP is to find the *shortest* $H$ (i.e., with the minimal number of atoms for easier interpretability) such that, denoting by $AS(P)$ the answer sets of an ASP program, we have:

$$\forall e \in E \ \exists as \in AS(B \cup H \cup C): \ e^{inc} \subseteq as, \ e^{exc} \nsubseteq as \tag{2}$$

ILASP also returns the number of examples that are not covered, if any. This is crucial when learning from data generated by stochastic processes, as traces of POMDP executions. In our scenario, partial interpretations contain atoms for `init` or `contd`, while the context involves environmental features.

### 3.4. Case studies

**Rocksample** In the rocksample domain, an agent can move in cardinal directions on a $N \times N$ grid, one cell at a time, with the goal of reaching and sampling a set of $M$ rocks with a known position on the grid. Rocks may be valuable or not. Sampling a valuable rock yields a positive reward ($+10$), while sampling a worthless rock yields a negative reward (-10). The observable state of the system is described by the current position of the agent and rocks, while the value of rocks is uncertain and constitutes the belief. It can be refined with a checking action, with accuracy depending on the distance between the rock and the agent. Finally, the agent obtains a positive reward ($+10$) exiting the grid from the right-hand side.

**Pocman** In the pocman domain, the pocman agent can move in the four cardinal directions (hence, $|A| = 4$) in a grid world with walls, to eat pellets of food (+1 reward). Each cell contains food pellets with probability $\rho_f$. $G$ ghost agents are also present, which normally move randomly, but may start chasing the pocman (with probability $\rho_g$) if they are close. If the pocman clears the level, i.e., it eats all food pellets, a positive reward is yielded (+1000). If the pocman is eaten or hits a wall, it receives a negative reward (-100). Moreover, at each step, the pocman gets a negative reward (-1). The fully observable state includes the positions of the walls and the agent. At each step, the pocman receives observations about ghosts and food within a 2-cell distance. Then, it builds two belief distributions, about the location of ghosts and food.

## 4. Methodology

We now describe our methodology for learning domain-related persistent macro-actions from POMDP executions and integrating them into MCTS-based planners to guide exploration. Starting from traces in the form of belief-action pairs $\langle b, a \rangle \in \mathcal{B} \times A$, we want to obtain a map $\Gamma : \mathcal{B} \to \mathcal{M}_A$, where $\mathcal{M}_A$ represents the set of macro-actions deriving from $\mathcal{A}$, i.e., the ASP formalization of $A$. $\Gamma$ is an ASP program containing EC theories to ground macro-actions from a given $b$, in combination with the POMDP transition model. $\Gamma$ can then be integrated in MCTS-based planners to guide exploration.

### 4.1. ASP formalization of the POMDP problem

First of all, we need to represent the POMDP domain (belief and action) through ASP terms. To this aim, we define a *feature map* $F_\mathcal{F} : \mathcal{B} \to \mathcal{H}(\mathcal{F})$ and an *action map* $F_\mathcal{A} : A \to \mathcal{H}(\mathcal{A})$, mapping collected beliefs and actions to a lifted logical representation, expressed as ground terms from $\mathcal{F}$. We require the following assumptions.

ASSUMPTION 1:
$\mathcal{A}, \mathcal{F}, F_\mathcal{F}$ and $F_\mathcal{A}$ are priorly known.

This is a common assumption, also made by Meli et al. (2024); Furelos-Blanco et al. (2021), and weaker than the availability of symbolic specifications or planners, as in Kokel et al. (2023). Indeed, $F_\mathcal{A}$ is a trivial symbolic re-definition of MDP actions. Moreover, we *do not require* $\mathcal{F}$ to be *complete*, i.e., to include all task-relevant predicates. Hence, we assume that basic predicates and their grounding ($F_\mathcal{F}$) can be defined with minimal domain knowledge, or they can be learned separately via automatic symbol grounding, as in Umili et al. (2023).

ASSUMPTION 2:
$F_\mathcal{A}$ is invertible.

This is realistic in most practical cases, since either a different predicate for each action can be defined (in case of discrete actions), or a simple transformation from macro-actions to actions can be introduced or learned, as shown by Umili et al. (2023). It is important to notice that, differently from Meli et al. (2024), we define $\mathcal{F}$ starting *only from the concepts represented in the POMDP transition map*, i.e., the effects of actions on the environment, excluding any other user-defined commonsense concept about the domain.

## 4.2. Macro-actions generation from execution traces

Traces of POMDP execution (belief-action pairs) represent sequences of specific instances of the task. After isolating the traces in which the agent obtained the highest return, we identify sequences of belief-action pairs where the action does not change: $\langle\langle b_1, \bar{a}\rangle, \ldots, \langle b_n, \bar{a}\rangle\rangle$, with $n > 1$. We then generate, for every $a \in A \setminus \{\bar{a}\}, 1 < i \leq n$, CDPIs in the form:

$$\langle\langle\{\mathtt{init}(F_{\mathcal{A}}(\bar{a}))\}, \{\mathtt{init}(F_{\mathcal{A}}(a))\}\rangle, F_{\mathcal{F}}(b_1)\rangle$$
$$\langle\langle\{\mathtt{contd}(F_{\mathcal{A}}(\bar{a}))\}, \{\mathtt{contd}(F_{\mathcal{A}}(a))\}\rangle, F_{\mathcal{F}}(b_i)\rangle$$

such that $e^{inc}$ contains observed groundings of $\mathtt{init}$ and $\mathtt{contd}$ for $\bar{a}$, while $e^{exc}$ contains groundings for unobserved actions. This allows us to learn only relevant EC theories about the task, i.e., axioms which generate macro-actions as in the traces of execution, according to Equation (2).

For instance, assume that the rocksample agent executes $\mathtt{east}$ twice to reach a valuable rock with id $2$ on its right (i.e. at distance $2$ on $x$ axis). The following CDPIs are generated[1]:

$$t = 1 : \langle\langle\{\mathtt{init(east)}\}, \{\mathtt{init(north)}, \mathtt{init(west)}, \ldots\}\rangle, \{\mathtt{dist(2,2)}, \mathtt{delta\_x(2,2)}, \ldots\}\rangle$$
$$t = 2 : \langle\langle\{\mathtt{contd(east)}\}, \{\mathtt{contd(north)}, \ldots\}\rangle, \{\mathtt{dist(2,1)}, \mathtt{delta\_x(2,1)}, \ldots\}\rangle \qquad (3)$$

At this point, the target ILASP hypothesis $H_a$ contains, $\forall a \in \mathcal{A}$, an EC theory for $a$. Combining $H_a$ with Equation (1) and the transition map $T_a$ for $\mathtt{east}$:

```
delta_x(R,D,t) :- delta_x(R,D-1,t-1),east(t-1).
```

we can generate the macro-action $M_a = \langle\mathtt{east(1)}, \mathtt{east(2)}\rangle$.
Formally, $\Gamma : \{\Gamma_a : \mathcal{B} \to M_a\}_{\forall a \in \mathcal{A}}$, and computing $\Gamma_a$ is equivalent to solving the ASP program $B \cup H_a \cup T_a \cup F_{\mathcal{F}}(b)$.

## 4.3. Integration of macro-actions as planning heuristics

Once $\Gamma$ is learnt, it can be integrated in POMCP and DESPOT (as well as in any extension of these algorithms) as shown in Algorithm 1. In POMCP, the goal is to guide UCT and rollout phases, generating macro-actions from $\Gamma$ which *suggest most promising sub-trees to be explored*, preserving optimality guarantees (Silver and Veness (2010)). In DESPOT, we want to reduce the gap $u(b) - l(b)$, providing a better approximation of $l(b)$ with macro-actions as *sequences of suggested default actions*.

We first compute macro-actions $M_a \: \forall a \in \mathcal{A}$ as explained above (line 1). At each time step $t$, in POMCP we push $t$-th action from each $M_a$ in list $A_H$ (line 8). We also initialize $N(ha) = N_{max}$[2] for UCT at line 9, making $V_{UCT}(ha)$ initially higher to encourage exploration of $a$. In rollout, we sample actions according to a weighted probability distribution $\pi(A, \rho)$, where weights $\rho$ are set at line 12 such that $\rho_a = cov_a$ if $a \in A_H$, $\rho_a = \min_{\mathcal{A}}\{cov_a\}$ if $a \notin A_H$. $cov_a$ is the coverage ratio for $a$, i.e., the percentage of not covered examples from $H_a$. The $\rho$ set is finally normalized. In this way, macro-actions have a higher probability of being selected. Notice that $\rho_a \neq 0 \: \forall a \in A$, hence, the *optimality guarantees of POMCP*

---

1. Since each CDPI corresponds to a specific time step, $\mathtt{t}$ is omitted for brevity.

2. Empirically set to 10 in this paper

---

**Algorithm 1:** POMDP planning with $\mathcal{M}_\mathcal{A}$

---

**Input:** $\{H_a\}_{\forall a \in A}$, POMDP, initial belief $b$, POMDP planner $alg$, $N_{\max} \in \mathbb{R}^+$

1   Initialize $\{M_a = \Gamma_a(b)\}_{\forall a \in A}, \quad t = 0, \quad stop \leftarrow$ False

2   **while** $\neg stop$ **do**

3      **if** $\exists a \in A : |M_a| > t$ **then**

4         $A_H \leftarrow []$;

5         **if** $alg == POMCP$ **then**

6            **foreach** $a \in A$ **do**

7               **if** $|M_a| > t$ **then**

8                  $A_H.\text{push}(M_a(t))$;

9                  $N(ha) = N_{\max}$;

10               **end**

11            **end**

12            $\rho \leftarrow \text{SET\_PROB}(A, A_H, \{cov_a\}_{a \in A})$;

13            $a' \leftarrow \text{ROLLOUT}(A, \rho, b)$ **or** $\text{UCT}(N(ha), b)$;

14         **end**

15         **else if** $alg == DESPOT$ **then**

16            Compute $u(b)$;

17            $\text{SORT}(\{M_a\}_{a \in A})$;

18            $A_H \leftarrow \text{POP}(\{M_a\}_{a \in A})$;

19            $l(b) \leftarrow Q(b, A_H(t))$;

20            $a' \leftarrow \text{SOLVE}(l(b), u(b))$;

21            $(b, stop) \leftarrow \text{STEP}(b, a')$;

22            $t \leftarrow t + 1$;

23         **end**

24         **end**

25         **else**

26            $t \leftarrow 0, \{M_a = \Gamma_a(b)\}_{\forall a \in A}$;

27         **end**

28   **end**

---

*are preserved*, including the *robustness to bad heuristics* (with a sufficiently high number of online simulations). In DESPOT, we sort $M_a$'s according to descending length (line 17); we then select the longest one and compute the lower bound as the $Q$-value of $a$ (lines 18-19).

Since selecting the longest macro-action allows exploiting the policy heuristic for the longest planning horizon, $\Gamma$ is only computed after the longest previous macro-actions have been used (line 26). For this reason, our approach is more efficient than Meli et al. (2024). Though the belief $b$ is updated meanwhile (line 21), MCTS still analyzes the updated belief (lines 16, 22). Moreover, differently from reward shaping, *we preserve Markovianity, since MCTS is only driven by temporal heuristics.*

## 5. Experimental results

We now present experimental results of applying our methodology to the rocksample and pocman domains[3]. For each of them, we first generate traces of execution, with POMCP endowed with $2^{15}$ online simulations and particles for belief approximation[4], in *simple set-*

---

3. For full reproducibility, we provide code at https://github.com/Isla-lab/persistent_macros_pomdp

4. Setting the number of simulations equal to the number of particles is a common design choice in POMCP. From now on, we will refer to *simulations* and *particles* interchangeably.

*tings*, e.g., small maps with few rocks in rocksample, or few ghosts in pocman. Among all traces, we select the ones with discounted return higher than the average and use them to generate macro-actions. We then evaluate the quality of the learned theories when integrated into MCTS-based planners. Specifically, we investigate the *robustness and utility* of the generated macro-actions when the number of online simulations is reduced in POMCP, the *generality* of the learned theories when the setting changes with respect to training conditions, and the *computational cost* of generating suggested macro-actions during online planning, measured as the time per step required by POMDP solvers. Appendix B also shows the comparison of our methodology with a recurrent autoencoder architecture proposed by Subramanian et al. (2022), showing that our method gains better results both in computational efficiency and in average discounted return and generalization.

### 5.1. Logical representations and learning results

We here detail how macro-actions were generated for rocksample and pocman. Learned EC theories, including coverage factors $\{cov_a\}_{\forall a \in \mathcal{A}}$, are reported only for pocman for brevity, while for rocksample they are reported in Appendix A.

**Rocksample**  For the rocksample domain, $\mathcal{F}$ contains: `dist(R,D)`, representing the 1-norm distance `D` between the agent and rock `R`; `delta_x(R,D)` (respectively, `delta_y(R,D)`), meaning $x$-(respectively, $y$-)coordinate of rock `R` with respect to the agent is `D`; `guess(R,V)`, representing the probability `V` that rock `R` is valuable; $X \leqslant \bar{x}$, where `X` is either `V,D` and $\bar{x}$ is a possible value for `X`. Each of these atoms is defined in the transition map, e.g., `delta_X(R,D)` is affected by moving actions and `guess(R,V)` by sensing. Action atoms are `east`, `west`, `north`, `check(R)` and `sample(R)`, however, since we learn from actions occurring in sequences of average length $> 1$, we only learn macro-actions for `north`, `east`, `south`, `west`. The training scenarios consist of $12 \times 12$ ($N = 12$) grids with 4 rocks ($M = 4$), with random positions and values.

**Pocman**  In pocman, $\mathcal{F}$ consists of the following environmental features: `food(C,D,V)` and `ghost(C,D,V)` representing the discrete (percentage) probability $V \in \{0, 10, \ldots, 100\}$ that a food pellet (or ghost) is within $D \in \mathbb{Z}$ Manhattan distance from the pocman in `C` cardinal direction, `C` $\in \{$`north, south, east, west`$\}$, and `wall(C)`, representing the presence of a wall in `C` cardinal direction. The set $\mathcal{A}$ of ASP actions consists of the single atom `move(C)`, being `C` $\in \{$`north, south, east, west`$\}$. The training scenario consists of a $10 \times 10$ grid with $G = 2$ ghosts that, when close enough to the pocman agent ($D < 4$), can chase it with $\rho_g = 75\%$ probability. Each non-wall cell may contain a food pellet with $\rho_f = 50\%$ probability. From this setting, we learn the following EC theory (with $cov = 75\%$ coverage factor):

```
init(move(C))  :- food(C,D1,V1), V1>30, D1<4,
                      ghost(C,D2,V2), V2<80, D2<6, not wall(C).
contd(move(C))  :- food(C,D1,V1), V1>30, D1<4,
                      ghost(C,D2,V2), V2<80, D2<6, not wall(C).
```

Thanks to the logical formalism, the above rules can be easily interpreted, and suggest the agent to move in directions with a moderate probability of finding ghosts ($< 80\%$), but a relatively high chance ($> 40\%$) of finding food pellets.

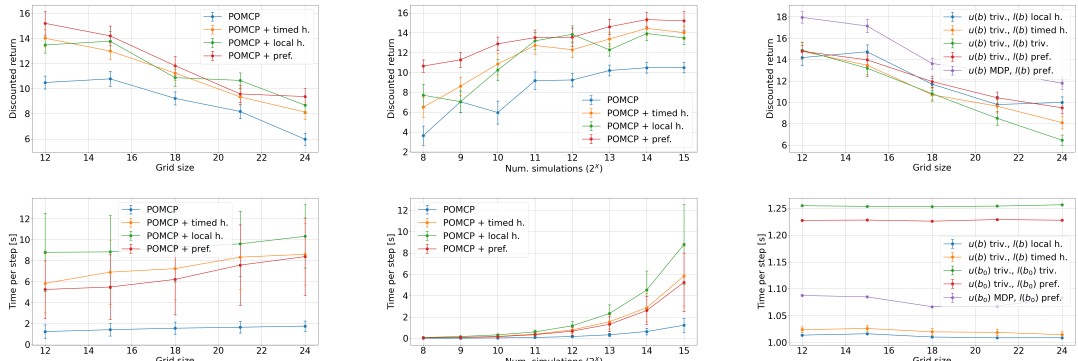

Figure 1: Planning performances (mean $\pm$ std) in rocksample, in POMCP varying $N$ (left) or the number of simulations (center), and DESPOT (right) varying $N$.

## 5.2. Online planning results

We now evaluate the performance of Alg. 1. We compare our methodology (*timed h.* in plots) against i) the time-independent heuristics learned by Meli et al. (2024)(*local h.* in plots); ii) handcrafted policy heuristics introduced by Silver and Veness (2010); Ye et al. (2017) (*pref.* in plots). Time-independent rules generate trivial macro-actions with $|M_a| = 1$, $\forall a \in \mathcal{A}$, hence requiring to recompute $\Gamma$ more often. Moreover, both time-independent and handcrafted heuristics contain more advanced concepts than the ones extracted from the transition map and used as environmental features, accounting, e.g., for the number of times an action has been executed [5].

**Rocksample** Figure 1 (left, center) shows POMCP performances in different challenging (with respect to the training setting) conditions, increasing $M$ to 8 and progressively incrementing $N$. Macro-actions (yellow curve) perform much better than pure POMCP (blue curve) and similarly to local heuristics (green curve) in terms of discounted return, almost reaching the performances gained exploiting handcrafted heuristics (red curve). Moreover, macro-actions are more convenient in terms of computational time per step. *This assesses the generality of our heuristics out of the training setting.* Following Ye et al. (2017), we consider two upper bounds for testing performances in DESPOT: *MDP*, which computes the expected value of a root node in MCTS via hindsight optimization, approximating the problem to a MDP; *trivial (triv.)*, where the maximum value of a node is set to $R_{max}/(1-\gamma)$, being $R_{max}$ the maximum task reward. We then evaluate *local h.* and *timed h.* as lower bounds, along with *triv.*, which defaults to `east` to exit. Figure 1 (right) shows that with $M = 8$, temporal (yellow), local (blue), and handcrafted (red) heuristics combined with the trivial lower bound yield similar returns, though lower than the best combination using a handcrafted lower bound and MDP upper bound. Furthermore, temporal heuristics are always more computationally convenient than the handcrafted ones with any upper bound in this domain with many actions. No significant discrepancy is evidenced between macro-actions and local heuristics. In fact, the computational performance of DESPOT depends purely on the quality of the heuristics, closely approximating the lower and upper bounds.

---

5. A detailed description of the heuristics for the rocksample domain can be found in Appendix A

Table 1: Pocman results with DESPOT (mean ± std) in nominal and challenging conditions

| | | Nominal conditions | | | | Challenging conditions | | | |
|---|---|---|---|---|---|---|---|---|---|
| | | $10 \times 10, G = 2$ | | $17 \times 19, G = 4$ | | $10 \times 10, G = 2$ | | $17 \times 19, G = 4$ | |
| $l(b)$ | $u(b)$ | Disc. ret. | Time/step [s] | Disc. ret. | Time/step [s] | Disc. ret. | Time/step [s] | Disc. ret. | Time/step [s] |
| Pref. | MDP | $52.37 \pm 3.52$ | $1.102 \pm 0.002$ | $72.56 \pm 2.04$ | $1.124 \pm 0.002$ | $12.98 \pm 3.07$ | $1.084 \pm 0.002$ | $16.30 \pm 1.58$ | $1.116 \pm 0.002$ |
| Timed h. | MDP | $\mathbf{59.02 \pm 3.64}$ | $1.104 \pm 0.002$ | $\mathbf{71.51 \pm 2.16}$ | $1.129 \pm 0.002$ | $\mathbf{21.21 \pm 4.70}$ | $1.096 \pm 0.002$ | $\mathbf{12.87 \pm 1.90}$ | $1.129 \pm 0.002$ |
| Local h. | MDP | $55.64 \pm 2.41$ | $1.092 \pm 0.002$ | $67.08 \pm 2.05$ | $1.109 \pm 0.002$ | $6.71 \pm 2.93$ | $1.075 \pm 0.002$ | $10.63 \pm 1.68$ | $1.093 \pm 0.002$ |
| Triv. | MDP | $35.56 \pm 3.34$ | $1.432 \pm 0.003$ | $21.78 \pm 3.30$ | $1.397 \pm 0.003$ | $9.29 \pm 4.33$ | $1.327 \pm 0.002$ | $-1.37 \pm 2.03$ | $1.303 \pm 0.002$ |
| Pref. | Triv. | $51.52 \pm 3.30$ | $1.105 \pm 0.002$ | $71.31 \pm 2.03$ | $1.119 \pm 0.002$ | $14.88 \pm 3.90$ | $1.105 \pm 0.002$ | $14.67 \pm 1.66$ | $1.119 \pm 0.002$ |
| Timed h. | Triv. | $\mathbf{63.26 \pm 3.24}$ | $1.099 \pm 0.002$ | $\mathbf{65.39 \pm 2.80}$ | $1.120 \pm 0.002$ | $7.57 \pm 2.63$ | $1.091 \pm 0.002$ | $9.78 \pm 1.44$ | $1.100 \pm 0.002$ |
| Local h. | Triv. | $53.42 \pm 3.32$ | $1.094 \pm 0.002$ | $64.10 \pm 2.37$ | $1.107 \pm 0.002$ | $9.40 \pm 1.89$ | $1.073 \pm 0.001$ | $11.07 \pm 1.52$ | $1.088 \pm 0.002$ |
| Triv. | Triv. | $2.67 \pm 3.13$ | $1.418 \pm 0.003$ | $-1.37 \pm 4.93$ | $1.350 \pm 0.003$ | $0.63 \pm 4.74$ | $1.305 \pm 0.002$ | $-4.84 \pm 2.82$ | $1.285 \pm 0.002$ |

**Pocman**   In pocman, we again consider more challenging settings, increasing the grid size (up to $17 \times 19$), the number of ghosts ($G = 4$) and their aggressiveness ($\rho_g = 100\%$), and reducing the number of food pellets ($\rho_f = 20\%$). This forces the agent to rely on truly general and robust heuristics to complete the task. In POMCP, temporal heuristics are more computationally efficient as for rocksample, especially on larger maps. However, given the complexity and very long planning horizon, even in the training setting, even handcrafted heuristics are hardly able to improve the discounted return, as shown in Appendix B.

Table 1 shows DESPOT performance in both training setting and challenging conditions. Here, the trivial lower bound suggests moving north, while the handcrafted (*pref.* in tables) one reads: *move in a direction where no ghost nor wall is seen, and avoid changing direction to minimize the number of steps.* This is similar to learned heuristics, but it does only account for observations (not beliefs) and ignores food. Moreover, it suggests not to change direction, which cannot be captured by our current logical formalization. Results about the computational time per step are analogous to rocksample; however, in terms of discounted return, macro-actions tend to perform better than local heuristics (see bold results in the tables). Moreover, they are sometimes able to significantly outperform even the handcrafted heuristics, even on the small map in challenging conditions. This shows the improved performance of our methodology on long-horizon tasks.

## 6. Conclusion

We showed an extension of POMDP logical action theory learning from example executions to account for time with EC, requiring minimal prior domain knowledge (only the POMDP transition model) if compared to previous works. The learned macro-actions can efficiently guide POMCP and DESPOT solvers, achieving extended expressivity with respect to time-independent heuristics and increasing the computational performance of POMCP. Moreover, our methodology better generalizes to unseen, more complex scenarios. We believe this is an important step towards learning temporal heuristics for planning under uncertainty and as a future work, we plan to extend the learning process to even more complex logical representations, such as LTL, also accounting for continuous domains. Moreover, we plan to validate our methodology in more challenging domains, e.g., robotics.

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

## Appendix A. Complete heuristics for the Rocksample domain

Here we provide the full EC theory learned for the rocksample domain:

```
init(east,t) :- V>70, D1<1, D2>0,delta_y(R,D1), delta_x(R,D2), guess(R,V).
init(west,t) :- V<90, D1<2, D2<0,dist(R,D1), delta_x(R,D2), guess(R,V).
init(south,t) :- V>60, D<0,delta_y(R,D), guess(R,V).
init(north,t) :- V>60, D1>1, D2>0,dist(R,D1), delta_y(R,D2), guess(R,V).
cont(east,t) :- V>70, D1<1, D2>0,delta_y(R,D1), delta_x(R,D2), guess(R,V).
cont(west,t) :- V<90, D1<2, D2<0,dist(R,D1), delta_x(R,D2), guess(R,V).
cont(south,t) :- V>60, D<0,delta_y(R,D), guess(R,V).
cont(north,t) :- V>50, D1<3, D2>0,dist(R,D1), delta_y(R,D2), guess(R,V).
```

All axioms correctly capture the directional constraint for moving towards a rock (e.g., east starts and continues until delta_x(R,D), D>0). The coverage ratios are $cov_{north} = 96\%$, $cov_{east} = 89\%$, $cov_{south} = 83\%$, $cov_{west} = 99\%$. Our methodology combines the above theory with axioms for check(R) and sample(R) from Meli et al. (2024), excluding dependencies from the subgoal atom target(R). The axioms from Meli et al. (2024), together with the coverage factors, are shown below (notice that the coverage factors are way lower than the ones obtained with our method).

```
east :- target(R), delta_x(R,D), D ≥ 1.
west :- target(R), delta_x(R,D), D ≤ -1.
north :- target(R), delta_y(R,D), D ≥ 1.
south :- target(R), delta_y(R,D), D ≤ -1.
0{ target(R): dist(R,D), D ≤ 1;
    target(R): guess(R,V), 70 ≤ V ≤ 80} M.
:∼ target(R), dist(R,D).[D@1, R, D]
:∼ target(R), min_dist(R), guess(R,V). [-V@2,R,V]
check(R) :- target(R), guess(R,V),V ≤ 50.
check(R) :- guess(R,V), dist(R,D),D ≤ 0, V ≤ 80.
sample(R) :- target(R), dist(R,D), D ≤ 0,guess(R,V), V ≥ 90.
```

with coverage factors *cov*: 65% for `north` and `south` actions, 57% for `east`, 73% for `west`, 85% for `check(R)` and 65% `sample(R)`. Finally, the handcrafted heuristic (*pref.* in plots) has the following interpretation: *either check a rock whenever the value probability is uncertain ($< 100\%$) and it has been measured few times ($< 5$); or sample a rock if the agent is at its location and collected observations are more positive (good value) than bad, or move towards a rock with more good than bad observations.*

## Appendix B. Additional experiments

### B.1. Experiments for pocman in POMCP

Figure 2 shows the performance of POMCP in pocman. In the caption, *nominal* refers to $\rho_f = 50\%, \rho_g = 75\%$, while challenging refers to $\rho_f = 20\%, \rho_g = 100\%$ The grid size is $17 \times 19$ and $G = 4$ ghosts are present. The results confirm the computational improvement as in rocksample, but the task is probably too hard for POMCP to gain benefit even from handcrafted heuristics.

### B.2. Comparison with neural architecture by Subramanian et al. (2022)

We now compare our methodology against the state-of-the-art solution proposed by Subramanian et al. (2022) (*AIS* in the plots). The authors train a reinforcement learning agent to solve POMDP problems, optimizing over the approximate information state, i.e., maximizing the information entropy as the policy space is explored. This is particularly helpful in large action spaces and exploits a recurrent autoencoder architecture to deal with long-horizon tasks, thus making rocksample the most interesting benchmark. Since the authors do not implement their methodology for pocman, we then only evaluate the rocksample domain with the original code. The agent can only generalize to different grid sizes, hence we train two agents[6] on a $12 \times 12$ grid, with $M = 4$ and $M = 8$, respectively. Our methodology proves superior in terms of *training efficiency*, since AIS training requires $\approx 1.5$ h and 20000

---

6. Training over 5 random seeds is reported in the supplementary.

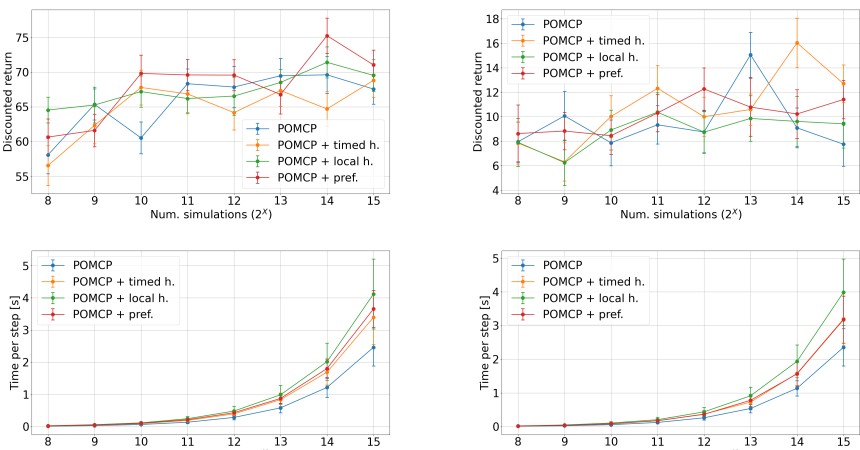

Figure 2: POMCP performances in the pocman scenario with $17 \times 19$ grid size and $G = 4$, both in nominal (left) and challenging (right) conditions.

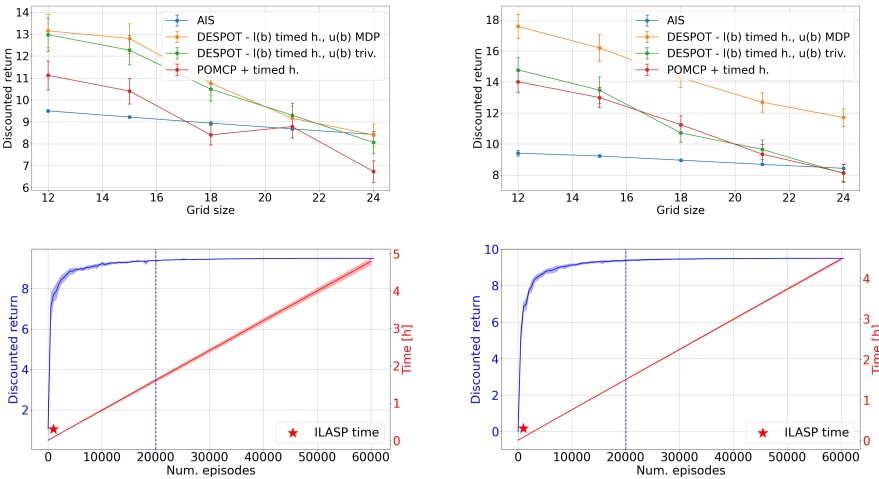

Figure 3: Comparison with AIS on rocksample with $M = 4$ (left) and $M = 8$ (right). Second row shows training results on the same settings.

episodes in both settings, while our methodology only 1000 traces (equivalent to episodes) within $\approx 18$ min. Furthermore, our approach is almost ever superior in terms of average discounted return and generalization, as shown in Figure 3. Figure 3 also shows the training performance of the neural architecture proposed by Subramanian et al. (2022), for $M = 4$ and $M = 8$ rocks. The mean and standard deviation over 5 random seeds are reported. The superiority of our method (star mark) is evident both in terms of data and time efficiency.

