# OpenReview forum: "Learning Symbolic Persistent Macro-Actions for POMDP Solving Over Time"
_nesyconf.org/NeSy/2025/Conference — NeSy 2025 Poster_

### Official Review · Reviewer_5iEG · 2025-03-23
**The authors learn event-calculus based ASPrules using ILASP to guide MCTS-based POMDP solvers**

**Rating:** 3
**Confidence:** 4

**Review:**

The authors learn event-calculus based logical rules using ILASP to guide MCTS-based POMDP solvers.  To do this, they lift beliefs and actions of the POMDP into a logical (i.e. ASP) representation which allows them to use traces of the POMDP to create ground facts used for ILP-based rule learning via ILASP and show that solving the resulting ASP program can act as a map from POMDP beliefs to macro-actions.  Experiments in two case studies show the method is competitive with hand-crafted heuristics or heuristics created in a more brute-force manner.

I found the paper interesting, generally well-written, and technically sound.  However, there is no neural component to the paper, which is odd as this is the neurosymbolic conference.  If we consider POMDP's to be "machine learning" (which is reasonable given the contributions to that area published in NeurIPS in recent years) then perhaps this can be considered as "informed machine learning" - but I the stated purpose of the conference is about "the integration of neural networks and symbolic AI."  This paper seems to be the integration of two symbolic methods: POMDP's and ILP.

**Anonymity:**

Remain anonymous

---

### Official Review · Reviewer_hvrK · 2025-04-04
**A method for improving MCTS + particle-filter based POMDP planning based on inferring event calculus theorems in ASP using ILP.**

**Rating:** 8
**Confidence:** 4

**Review:**

This work proposes a method for improving interpretability, computational efficiency, and the amount of necessary domain knowledge injection for planning in POMDPs. The work builds on top of Meli et al. (2024), which seems to have introduced the framework presented in the paper. Succinctly, Meli et al. (2024) propose to guide a POMDP planner based on MCTS and a particle filter by solving domain-specific ASP problems, where the body of an ASP axiom corresponds to the belief about the environment state, and the head is an admissible action given the state. Inductive Logic Programming is used to infer ASP rules given background knowledge, an ASP axiom search space, as well as a set of examples in the described CDPI form.

This work extends the set of inferable ASP axioms to a subset of event calculus theories. The EC theories have a temporal element to them, stating that certain actions should be executed for as long as the belief about the POMDP state fulfils certain conditions. Apart from reducing the cost of the MCTS method by proposing longer sequences of actions, EC theories may also increase the interpretability of the plan.

The work is technically very interesting. With few exceptions such as the missing line numbers in the algorithm, the writing is generally of very high quality. The experimental validation confirms that the proposed method reduces the algorithm runtime across various configurations while preserving the accuracy of previous approaches.

On the other hand, I believe the paper could benefit from a more detailed exposition, perhaps a sketch of the approach. The method is somewhat difficult to understand due to the large amount of notation introduced. While the notation is itself clear and consistent, it would be helpful to at least have comments in the algorithm pseudocode.

**Anonymity:**

Remain anonymous

---

### Official Review · Reviewer_pvvT · 2025-04-05
**The paper presents promising advancements in combining symbolic reasoning with probabilistic planning for POMDPs, addressing critical challenges like interpretability and computational efficiency. Its contributions are substantial and relevant to the field of neurosymbolic reasoning.**

**Rating:** 7
**Confidence:** 3

**Review:**

Summary:
The paper presents a novel methodology for integrating symbolic reasoning with POMDPs to improve decision-making under uncertainty. The authors propose using Inductive Logic Programming to learn persistent macro-actions represented in Event Calculus formalism, which are then used to guide MCTS-based POMDP solvers. This approach eliminates the need for handcrafted heuristics and enhances computational efficiency while maintaining interpretability. Experimental results on benchmark domains such as Rocksample and Pocman demonstrate the effectiveness of the learned macro-actions in improving planning performance and generalizing to complex scenarios.

Strengths:
1. The integration of ILP and EC to learn macro-actions is a significant contribution, as it bridges the gap between symbolic reasoning and probabilistic planning. This approach enhances interpretability, which is often lacking in black-box methods like neural networks.

2. Reduction in Computational Complexity: The paper shows that learned macro-actions improve computational efficiency in POMCP and DESPOT solvers by reducing inference time and guiding exploration effectively.

3. Robustness and Generalization: The experimental results demonstrate that the proposed methodology generalizes well to more complex scenarios, outperforming or matching handcrafted heuristics in terms of discounted return and computational time.

4. The methodology relies only on the POMDP transition model, making it less dependent on extensive prior knowledge compared to other approaches.

5. The paper provides thorough experimental validation across two benchmark domains with varying levels of complexity, showcasing the robustness of the proposed approach.

Limitations:
1. Dependence on Quality of Execution Traces: The quality of learned macro-actions heavily depends on the traces selected for training. Although ILP is robust to noisy data, bad traces could still impact the performance. However the criteria for selection of traces seem natural, and it could be interesting to see if it can be optimized further.

2. Expanding more on use of LTL could be a possible direction of future research.

3. Scalability Concerns for real-world applications with more complex action and state spaces remain to be explored.

4. **Clarity in Presentation**: Some sections, particularly those describing methodology (e.g., ASP formalization and macro-action generation), are dense and could benefit from clearer explanations or visual aids to improve accessibility for readers unfamiliar with ASP or EC.

Comment: Strongly recommend adding more visual aids (graphics).

**Anonymity:**

Remain anonymous

---

### Official Review · Reviewer_d8P8 · 2025-04-14
**A sound integration of Event Calculus and ILP for macro-action learning, with empirical results but some liminitation in the theoretical guarantees**

**Rating:** 6
**Confidence:** 4

**Review:**

The paper is overall clearly written, structured effectively, and easy to follow. In particular, the main contributions of the paper can be summarized as follows:

1- The methodological integration of Event Calculus with Inductive Logic Programming is sound and clearly described. Leveraging EC axioms allows temporal persistence, which is crucial for macro-actions effectiveness.

2- the technical approach to generating persistent macro-actions from successful execution traces using CDPIs is both elegant and novel. It efficiently encapsulates sequential decisions, outperforming simpler heuristics.

3 - The technical experiments demonstrate both generalization and computational advantages across benchmark tasks.  The show that learned macro-actions improve computational efficiency compared to traditional heuristics, especially in conditions with limited computational budgets.

My main concers are:

- The paper lacks rigorous theoretical analysis or formal guarantees about optimality, convergence, or bounds of the heuristic-driven MCTS exploration. While MCTS methods like POMCP have known optimality guarantees under sufficient exploration, introducing macro-actions and modifying rollout policies can affect these properties.
- The assumption (Assumption 1 and 2 in Section 4.1) regarding the invertibility and straightforwardness of symbolic grounding  is somewhat restrictive in my view. Real-world sensor data may not trivially yield symbolic predicates, and a discussion or strategy addressing cases where such grounding is challenging or ambiguous would be beneficial.
- Macro-actions learned from ILP might tend to become overly specialized to the traces used for training.
- The authors rely on coverage metrics (cov) to quantify the effectiveness of learned heuristics. However, coverage alone may not sufficiently capture heuristic quality, as it ignores action-value or reward-driven considerations. Beyond coverage (cov), they should propose or include additional metrics to measure heuristic effectiveness quantitatively, such as expected discounted return from partial heuristic-driven policies or robustness metrics under stochastic conditions.


Minor commens:

- "TO" this aim is repeted in Section 2 and Section 3.2
- The training scenarios consist of 12 × 12 (N = 12) grids with M = 4 rocks, with random positions and values. -> ... with four grids (M=4) ...
- though the belief b is updated in the meanwhile -> .. is updated meanwhile.
- macro-actions have similar return, though lower compared to the best combination with handcrafted lower bound and MDP upper bound. -> macro-actions achieve similar returns, though lower than the best-performing combination of handcrafted lower bound with MDP upper bound.
- From this setting, learn the following EC theory -> ..., we learn the ...

**Anonymity:**

Remain anonymous